# HYBRID SPATIAL REPRESENTATIONS
# FOR SPECIES DISTRIBUTION MODELING

## ABSTRACT

We address an important problem in ecology called Species Distribution Modeling (SDM), whose goal is to predict whether a species exists at a certain position on Earth. In particular, we tackle a challenging version of this task, where we learn from presence-only data in a community-sourced dataset, model a large number of species simultaneously, and do not use any additional environmental information. Previous work has used neural implicit representations to construct models that achieve promising results. However, implicit representations often generate predictions of limited spatial precision. We attribute this limitation to their inherently global formulation and inability to effectively capture local feature variations. This issue is especially pronounced with presence-only data and a large number of species. To address this, we propose a hybrid embedding scheme that combines both implicit and explicit embeddings. Specifically, the explicit embedding is implemented with a multiresolution hashgrid, enabling our models to better capture local information. Experiments demonstrate that our results exceed other works by a large margin on various standard benchmarks, and that the hybrid representation is better than both purely implicit and explicit ones. Qualitative visualizations and comprehensive ablation studies reveal that our hybrid representation successfully addresses the two main challenges. Our code is open-sourced at `https://anonymous.4open.science/r/HSR-SDM-7360`.

## 1 INTRODUCTION

Understanding species distribution ranges is a key issue in ecological research, and it has become increasingly important in the context of the current global climate crisis and biodiversity decline. Conventionally, species distribution data has been collected through field studies by human experts and explorers, who must gather and assess large amounts of information to determine whether a species is present in a given region. These processes are typically slow and labor-intensive, and by the time the models are completed, they may already be outdated or irrelevant.

Species Distribution Modeling (SDM) is a method that uses collected data to directly predict the distribution range of species, thus making related ecological research easier (Elith & Leathwick, 2009; Elith et al., 2010; Miller, 2010), and is widely applied in fields such as climate change assessment (Santini et al., 2021), invasive species management (Srivastava et al., 2019), and extinction risk mapping (Ramirez-Reyes et al., 2021). Whilst such models have achieved some success over the past two decades, most SDMs remain poor indicators of important ecological parameters (Lee-Yaw et al., 2022). Consequently, new SDM methodologies employing more advanced modeling techniques have continued to emerge (Beery et al., 2021).

One challenge in constructing SDMs is the collection of sufficient data for both training and testing (Feeley & Silman, 2011; Vaughan & Ormerod, 2005), as well as ensuring data quality (Hartig et al., 2024). For example, due to the nature of the data collection process, most large species distribution datasets are highly susceptible to sampling bias, class imbalance, and noise (Benkendorf et al., 2023; Dubos et al., 2022; Kramer-Schadt et al., 2013). Recent advances (Cole et al., 2023) in using deep learning for SDMs has reduced the demand for large amounts of high-quality data. In particular, some recent methods applying implicit neural representations achieved considerable accuracy, and no longer required training signals besides presence-only data. However, in practice, predictions from those models are often of limited spatial precision due to the implicit nature of

Figure 1: Results of Independent Component Analysis (ICA) on the feature embeddings for implicit, explicit, and hybrid models. The explicit embedding captures higher-frequency information and reflects local environmental data, while the implicit embedding, as a global location encoder, is less noisy. The hybrid representation combines the strengths of both. Note the noise in the explicit embeddings are mainly caused by the presence-only and community-sourced natures of our training data used.

their representational schemes: neural networks inherently produce global embeddings that are not grounded in local features.

In this paper, we explore a challenging task that highlights the limitations of implicit representations. First, we use **presence-only data** instead of presence-absence data. Since confirming a species' presence is generally easier than confirming its absence, many previous studies have constructed SDMs using presence-only data (Barbet-Massin et al., 2012; Mac Aodha et al., 2019; Cole et al., 2023), making this a more difficult but valuable task. Second, we use **no additional environmental information**. Although conventional SDMs usually use a lot of environmental inputs, and satellite images are also a common source of information (Dollinger et al., 2024; Gillespie et al., 2024; He et al., 2015; Klemmer et al., 2023), those data are notoriously difficult to obtain and clean, and are usually noisy — hence we will focus on exploring the locational embedding and thus not use those information. Third, similar to most previous deep learning-based methods, we use iNaturalist, a **community-sourced dataset**, which, as previously discussed, presents various difficulties. Finally, we construct a single model for **a large number of species simultaneously**.

Tailored to such challenges, inspired by advances in explicit and hybrid representations, we propose a **Hybrid Spatial Representation** for Species Distribution Modeling. Our representation combines an implicit component based on FCNet (Mac Aodha et al., 2019) with an explicit component based on multiresolution hashgrids (Müller et al., 2022), forming a hybrid model well-suited for the SDM task. An intuitive visual juxtaposition of these embeddings is shown in Figure 1. Experiments show that our method achieves the best of both worlds, producing state-of-the-art results on this challenging task. We also investigate the mechanisms behind the effectiveness of hybrid representations and characterize our model across a wide range of settings and evaluation methods for additional insights.

## 2 RELATED WORKS

**Species Distribution Modeling**  As discussed earlier, SDM is a challenging field that often requires learning from large volumes of inaccurate data. Recently, several works have used deep learning (Botella et al., 2018; Chen et al., 2017; Cole et al., 2023; Mac Aodha et al., 2019) to create SDMs from those massive datasets. This is a difficult task due to the inherent difficulties in the data, and therefore requires highly effective representational methods. The current state-of-the-art representation, as verified by several works (Cole et al., 2023; Lange et al., 2024; Rußwurm et al., 2023), is the FCNet architecture (Mac Aodha et al., 2019), which makes use of a Residual Network (He et al., 2016)-based structure to achieve effective implicit location embeddings.

It is understandable that explicit or hybrid representations have not been previously applied in SDMs, since — as our experiments will later demonstrate — explicit representations often produce noisy predictions with artifacts. However, there is a strong rationale for using explicit representations in SDMs: popular implicit embedding-based models, which rely on neural networks, struggle to capture local details. Our work demonstrates the power of combining implicit and explicit representations for SDM construction, and achieves state-of-the-art results on challenging benchmarks.

Note that while many current works use environmental information to supplement their models, our choice of not pre-assuming the use of environmental information (although we do conduct an experiment on it) is intended to narrow the problem frame. As prior works have highlighted, constructing

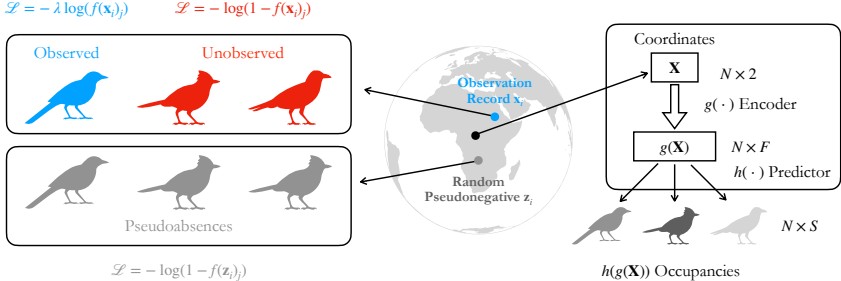

Figure 2: Illustration of our problem formulation and basic model structure.

expert range maps from observations is fundamentally distinct from inferring species presence using local environmental parameters. While environmental parameters may be helpful to the former, choosing effective ones requires extensive expertise. SDM works (*e.g.*, ones surveyed in Lee-Yaw et al. (2022)) often collect their own *ad hoc* environmental datasets. Hence, we choose to focus the paper on the theoretical and foundational merits of explicit and hybrid representations.

**Implicit, Explicit, and Hybrid Representations**  In fields such as signal processing, 3D vision, and computer graphics, Neural Implicit Representations (NIR) have achieved great results (Sitzmann et al., 2020; Mildenhall et al., 2021). A common pattern in those fields is that the success of implicit representations was often followed by the appearance of explicit ones which offer advantages such as increased accuracy and efficiency (Chen et al., 2022; Sun et al., 2022; Yu et al., 2021). In practice, hybrid representations that combine both implicit and explicit schemes often achieve superior performance by capturing the strengths of of both approaches.

While some previous works in the context of global spatial encoding have used hybrid or explicit representations (Kim et al., 2024; Mai et al., 2020; Rußwurm et al., 2023), to the best of our knowledge, no works on SDM have done this. In this work, we design an explicit representation specifically suited to the task of SDM construction based on multiresolution hashgrid representation. We demonstrate the merits of explicit representations, and show that our hybrid representation outperforms both implicit and explicit schemes by combining their advantages.

## 3 METHODS

### 3.1 PRELIMINARIES

Let $\mathcal{P}(\cdot) : [-1, 1]^{n \times 2} \to \{0, 1\}^{n \times S}$ be the ground-truth presence function, where $S$ is the number of species in the model, and for the coordinates $(lat, lon)$ (regularized between -1 and 1), we have $\mathcal{P}([lat, lon])_i = 1$ iff the species $i$ is present at $(lat, lon)$. Let $\mathbf{X} \in [-1, 1]^{N \times 2}$ be a matrix of coordinates where observations have been performed, where $N$ is the number of observation entries. Corresponding to the observations are species indices $\mathbf{s} \in [1, S]^N$, where $\mathcal{P}(\mathbf{x}_n)_i = 1$ if $i = s_n$. Here $\mathbf{x}_n$ represents the $n$th row of $\mathbf{X}$. While it is possible in practice that false presences occur through misidentification or aberrant migration, we treat this as regular noise in the data rather than a part of the problem formulation. Hence we need to construct a model $f(\cdot) : [-1, 1]^{n \times 2} \to \{0, 1\}^{n \times S}$ such that $f(\mathbf{X})$ approximates $\mathcal{P}(\mathbf{X})$.

Given random all-unlabeled pseudoabsences $\mathbf{Z} \in [-1, 1]^{N \times 2}$, the Assume Negative Full Loss as inCole et al. (2023) can be written as follows:

$$\mathcal{L}_{\text{full}}(\mathbf{X}, \mathbf{s}, \mathbf{Z}) = -\frac{1}{NS} \sum_{i=1}^{N} \sum_{j=1}^{S} (\mathbb{1}_{j=s_i} \lambda \log f(\mathbf{x}_i)_j + \mathbb{1}_{j \neq s_i} \log(1 - f(\mathbf{x}_i)_j) + \log(1 - f(\mathbf{z}_i)_j)), \quad (1)$$

where subscripts represent row slices, and $\lambda$ is a hyper-parameter to prevent the latter two terms from dominating.

Under the NIR-based setting, the function $f(\mathbf{X})$ consists of two parts: a location embedding $g(\cdot) : [-1, 1]^{N \times 2} \to \mathbb{R}^{N \times F}$ (where $F$ is the dimension of the embedding, also known as the number of features), and the occupancy predictor $h(\cdot) : \mathbb{R}^{N \times F} \to \mathbb{R}^{N \times S}$ which takes the embeddings as input and outputs the occupancy predictions for the species. Hence the model is described by:

$$f(\mathbf{X}) = h(g(\mathbf{X})). \quad (2)$$

Usually $h(\cdot)$ has a simple structure, such as being a single linear layer, while $g(\mathbf{X})$ is a more sophisticated model. Hence the significance of the embedding $g$ lies in encoding the coordinates in such a

Figure 3: Illustration of our multiresolution hashgrid representation's mechanism. The explicit embedding of any position (cyan) is calculated via calculating features (green) at each resolution level (blue) using bilinear interpolation and concatenating them.

manner that it would be easy for $h$ to conduct the final mapping step. We call the architecture of $g$ the *representation scheme*. The current state-of-the-art is FCNet (Mac Aodha et al., 2019), a design based on residual blocks.

At the beginning of $g$ there is also often a positional encoding of the coordinates, as perSitzmann et al. (2020), implicit representations perform better when the inputs are of high frequency. FCNet uses the wrap encoding $(\sin(\pi lon), \cos(\pi lon), \sin(\pi lat), \cos(\pi lat))$. Figure 2 displays a brief summary of the data and model structure in our problem formulation.

### 3.2 MOTIVATION

We notice two insufficiencies in the implicit formulation above.

**Global Parameterization** Since $g$ is a neural network, in the back-propagation process, most parameters have nonzero gradient steps. More intuitively, one can see the implication that each parameter is equally capable of being associated with the Amazon Rainforest as with the Saharan Desert. In the process of training parameters gradually get implicitly mapped to different features, but there is still no guarantee that the parameters can reliably describe local environmental information.

**Low Signal Frequency** In addition, since MLPs follow the Lipschitz constraint, intuitively maintaining a degree of "smoothness," they often struggle to describe high-frequency patterns. Indeed, a lot of previous work in NIRs has focused on encoding the data to facilitate modeling. In our task, this means that embeddings for nearby locations tend to be similar regardless of their characteristics, which indicates an inability to describe local details. In practice this is very undesirable, since many ecological boundaries cause sharp distinctions between ecosystems in physical proximity of each other.

**Principle for Explicit Embedding** We propose a guiding principle which could simultaneously solve to problems above. We introduce explainable parameters which each correspond to only a specific region on Earth. If a data point is not contained within the region associated with a parameter, the gradient of that parameter with respect to this input is zero, thus creating *local* instead of *global* parameterization. In addition, the Lipschitz constraint no longer applies at grid region boundaries, and thus the output can have arbitrarily high signal frequency. Such a model would require an explicit rule dictating the correspondence between parameters and geographical regions, which no prior work has considered. Hence, we introduce a new explicit embedding scheme which suits our purposes.

### 3.3 HYBRID SPATIAL REPRESENTATION

We propose using a multiresolution hashgrid encoding scheme as an explicit representation for SDM modeling. Specifically, we divide the Earth's surface into multiple grids with different resolution levels, and store trainable feature parameters associated with lattices of the grid in a hashtable. Embedded features on any given point for each resolution level are then calculated via bilinear interpolation. Finally, the output embedding is given by concatenating all hashed features from the different layers. An intuitive depiction is shown in Figure 3.

The resolution of each layer follows a geometric progression. Given maximum and minimum resolutions $R_{\max}$ and $R_{\min}$, and a total of $L$ layers, the resolution of layer $l$ is calculated as follows:

$$R_l = R_{\min} \exp\left(\frac{l}{L-1}(\log R_{\max} - \log R_{\min})\right).$$ (3)

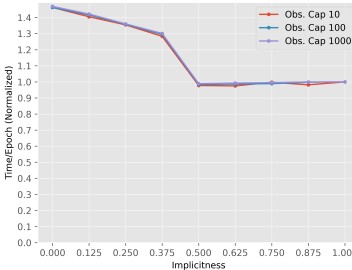
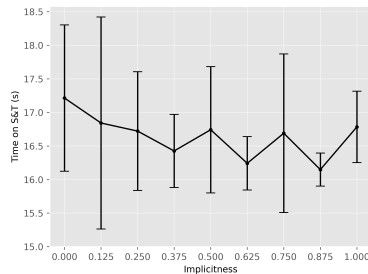

Figure 4: The training time per epoch of different models, divided by the training time of the implicit model for normalization.

Figure 5: The inference time of models on the S&T baseline. Error bars are $\pm$ one standard deviation across five repetitions.

The grids allow for our model to explicitly capture information regarding the local environment by using the features as representation. Furthermore, the different resolutions allow for the description of explicit interactions in between smaller grids, based on their mutual intersection with larger grids. The hashgrid, meanwhile, ensures that the number of parameters created is not of overly large size.

We then aggregate the locational embeddings from two parallel location encoders, one using the conventional implicit scheme and one using our explicit multiresolution hashgrid. We concatenate resulting embeddings from the two, and proceed to input the concatenated results as the embedding for the occupancy predictor.

Letting $g_i(\cdot) : [-1, 1]^{N \times 2} \to \mathbb{R}^{N \times F_i}$ and $g_e(\cdot) : [-1, 1]^{N \times 2} \to \mathbb{R}^{N \times (F - F_i)}$ represent the encoders, the resulting hybrid model is:

$$f(\mathbf{X}) = h(g_i(\mathbf{X}) \oplus g_e(\mathbf{X})), \tag{4}$$

where $\oplus$ represents concatenation along the second dimension, and $F_i$ is the dimension of the implicit embedding. We refer to the ratio $\frac{F_i}{F}$ as "implicitness," and treat it as a tunable hyper-parameter. For an explicit model with $L$ layers and $M$ features per level, we have $F - F_i = L \times M$. In practice we keep $M$ to powers of two for implementational reasons and vary $L$ for tuning implicitness.

## 4 EXPERIMENTS

### 4.1 BENCHMARKING

The dataset used was iNaturalist, which is a popular choice for benchmarking SDMs and has been used in multiple previous works (Mac Aodha et al., 2019; Cole et al., 2023; Rußwurm et al., 2023). We use standardized settings (Cole et al., 2023) in order to ensure fair comparison. To deal with class imbalance, as well as to condition on the amount of data provided, we sample 10, 100, or 1000 observations per species from the dataset, referred to below as the "Observation Cap" or "Obs. Cap" for short. The data, and thus the model, covers a total of 47375 species. It should be noted that the purely implicit version of our model, which uses FCNet (Mac Aodha et al., 2019), is architecturally congruent to SINR (Cole et al., 2023). More details regarding our settings can be found in the Appendices.

**Comparison on SDM Benchmarks** We evaluate our models on two human expert-created distribution range datasets: S&T (eBird Status and Trends) and IUCN (the International Union for Conservation of Nature). There are a total of respectively 535 and 2418 species overlapping between the iNaturalist training dataset and the two testing datasets, and we report the Mean Average Precision (mAP) of the models' predictions. Results are shown in Table 1. The reported results for all our models are means from five repetitions: for error bars please refer to Subsection 4.2, and for the full raw data please refer to Appendix A.

As shown, our models consistently achieve state-of-the-art results on those standardized tasks, by margins of up to **21.2%** relative improvement (for few-shot learning with 10 samples/species on the difficult IUCN benchmark), demonstrating the benefit of using a hybrid representation. In addition, we see that the model with implicitness 0.5 performs well across all scenarios, ruling out the need for extensive tuning of the implicitness hyper-parameter in practice (discussed more in Subsection 4.2).

Table 1: Results of experiments on S&T and IUCN benchmarks. Reported values are mAP percentages. Values in parentheses represent implicitness. Values for SINR, GP, and BDS are as reported by Cole et al. (2023). Results for BDS used all training data with no observation cap. We achieve large improvements compared to previous works, especially on the difficult IUCN task.

| Benchmark | S&T | | | IUCN | | |
|---|---|---|---|---|---|---|
| Obs. Cap | 10 | 100 | 1000 | 10 | 100 | 1000 |
| Ours-Explicit (0.0) | 60.21 | 71.23 | 76.01 | 48.83 | 62.46 | 64.23 |
| Ours-Hybrid (0.25) | 66.76 | 75.05 | 77.86 | 58.30 | 69.02 | 68.03 |
| Ours-Hybrid (0.5) | 66.64 | 75.27 | 78.47 | 59.39 | 69.57 | 70.32 |
| Ours-Hybrid (0.75) | 66.54 | 75.01 | 78.01 | 58.28 | 69.23 | 69.46 |
| Implicit (1.0) | 65.59 | 73.12 | 76.81 | 50.98 | 62.06 | 65.57 |
| SINR (Cole et al., 2023) | 65.36 | 72.82 | 77.15 | 49.02 | 62.00 | 65.84 |
| GP (Mac Aodha et al., 2019) | | | 73.14 | | | 59.51 |
| BDS (Berg et al., 2014) | | | 61.56* | | | 37.13* |

Table 2: Results of experiments on the GeoFeature benchmark. Reported values are averaged $R^2$ correlations across eight environmental features. Values for SINR and GP are as reported by Cole et al. (2023). As shown, explicit models are the most correlated with environmental features.

| | Ours (Implicitness) | | | | | SINR | GP |
|---|---|---|---|---|---|---|---|
| Obs. Cap | 0.0 | 0.25 | 0.5 | 0.75 | 1.0 | | |
| 10 | 74.6 | 74.5 | 73.5 | 72.5 | 71.1 | 71.2 | |
| 100 | 78.0 | 78.0 | 77.1 | 76.6 | 73.9 | 73.6 | |
| 1000 | 79.3 | 79.0 | 78.6 | 77.9 | 75.2 | 75.2 | 72.4 |

**Correlation with Environmental Data**   We investigate whether our explicit representation indeed represents local environmental information better as expected by using some common environmental parameters as proxies. The data comes from the GeoFeature benchmark (Cole et al., 2023), and includes 8 parameters in different locations sampled within the contiguous United States, such as above-ground carbon, elevation, *etc*. We report the average $R^2$ correlation between the embeddings and the environmental parameter. Results are shown in Table 2.

As shown, explicit models are the most correlated with environmental information, as we expected in our design. There is also a very clear negative relation between implicitness and the performance on this task. We conclude that explicit models have strong capability for capturing environmental information.

**Training and Inference Speed**   We trained models for all 9 implicitness settings for a single epoch on single RTX A4000 (16GB) GPUs, running five repetitions simultaneously. We then conduct inference under the same settings on the S&T benchmark. We found that all standard deviations for the training time are within 2% times the mean, suggesting that the results have high statistical significance, so we omit reporting them and just report the means here. Results are shown in Figures 4 and 5.

As shown, we see that when implicitness is less than 0.5 (the model leans explicit), there seems to be a training overhead of up to around 47% times the implicit model, presumably due to the larger number of features per level. However, for models with implicitness greater than or equal to 0.5 (the model leans implicit), there is no overhead compared to the implicit model. Hence, using our hybrid model for better results does not require incurring sacrifices in speed. Meanwhile, no statistically significant difference between the models was observed for inference time.

**Comparison with Larger Implicit Networks**   In all of the experiments above, the dimension of the location embedding is 256 as in SINR. One may wonder whether simply increasing the di-

Table 3: All were run with an observation cap of 1000. The hybrid model had implicitness 0.5. Results display that simply increasing the dimension of the implicit embedding (and thus the number of parameters) does not result in results comparable to our hybrid method, while it *does* cause excruciating computational overhead.

| | Number of Features (Implicit) | | | | Ours (Hybrid) |
|---|---|---|---|---|---|
| | 256 | 512 | 1024 | 2048 | |
| Training Time (s) | 574 | 839 | 1376 | 2760 | 568 |
| Inference Time (s) | 16.8 | 23.2 | 24.7 | 27.7 | 16.7 |
| S&T mAP (%) | 77.15 | 77.73 | 78.02 | 78.29 | 78.47 |

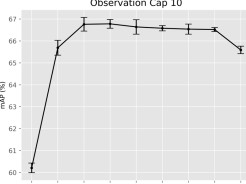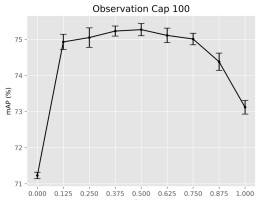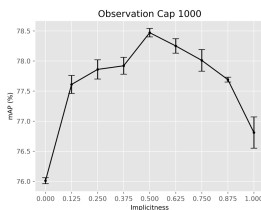

Figure 6: Results of models with different implicitness on the S&T task. Error bars are ± one standard deviation across five repetitions. As shown, implicitness is not a very sensitive hyper-parameter.

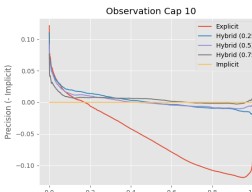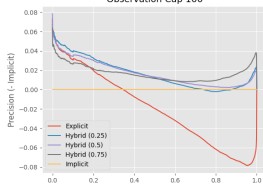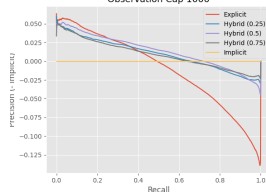

Figure 7: Precision-Recall Curves (regularized by subtracting the implicit curve) averaged across the 535 species on the S&T benchmark. Precision is regularized with respect to the implicit models. As shown, models with lower implicitness have higher precision in low-recall scenarios, and vice versa. The significant drop in high-recall precision for explicit models can be attributed to tendency of overfitting.

mension of this feature embedding would allow the resulting "fat" implicit model to achieve better results than our explicit representation. We conduct experiments to show that this is not the case: while marginal performance gains can be achieved, they come at a very heavy cost for training and inference speed, and still cannot exceed results of our hybrid model. Results are reported in Table 3.

## 4.2 CHARACTERIZATION

**Hyper-Parameter Sensitivity Analysis of Implicitness**   Here we present data from all 9 implicitness settings under the two benchmarks, each ran for five repetitions to rule out the effect of randomness. All reported results are under the best learning rate settings for respective models. Results are displayed in Figure 6.

Our results further verify that all values of implicitness except 0.0 and 1.0 (the degenerate cases) are relatively insensitive and robust: all hybrid models have several standard deviations' improvement compared to explicit or implicit ones. Hence, no extensive tuning is needed for this newly introduced hyper-parameter. In practice, we recommend a simple value of 0.5, which is also the most efficient computation-wise.

**Precision-Recall Trade-Off**   To further identify mechanisms via which the hybrid model achieves superior results, we plot the Precision-Recall Curves (PRCs) for models with 0.0, 0.25, 0.5, 0.75, and 1.0 implicitness here. We use PRCs instead of other tools like ROCs because it is better suited to our task, which has strong class imbalance (Saito & Rehmsmeier, 2015). Results are shown in Figure 7.

It can be seen that the improvement our models achieve stem mainly from their high precision for low-recall scenarios in comparison to implicit models. However, the precision of explicit models plunge when recall is high due to overfitting. Hybrid models successfully balance between the two, achieving the best of both worlds.

**Conditioning on the Number of Species**   To verify that hybrid and explicit models are better at aggregating information from the distribution of multiple species, we run the S&T and environmental data baselines again with different numbers of species. Following (Cole et al., 2023)'s approach, we train the models on the 535 S&T species only first, and increment the number of species in intervals of 4000. All models in this experiment were trained with an observation cap of 1000 observations per species. Results are shown in Figures 8 and 9.

We can notice from the results that on the S&T benchmark, hybrid models perform better than others, and that this gap generally tends to grow as the number of species increases, suggesting

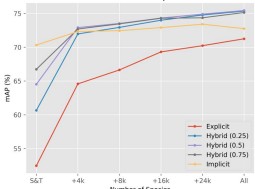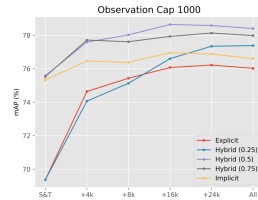

Figure 8: Results on the S&T benchmark with respect to the number of species. Hybrid models experience larger increases in performance as the number of species increase. Again note trends of overfitting for the explicit model.

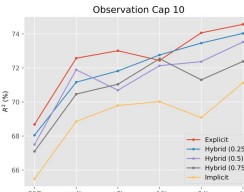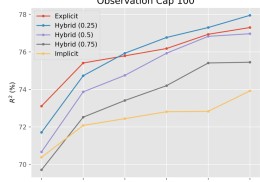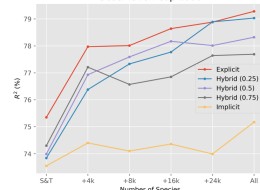

Figure 9: Results on the GeoFeatures benchmark with respect to the number of species. Explicit models experience larger increases in performance as the number of species increase, due to their inherent ability of inferencing environmental information from species observations.

the superiority of hybrid models in aggregating information across species. The implicit baseline, meanwhile, learns only a limited amount of new information from having more species modeled. Our experiments on correlation with environmental data further show that explicit models are very good at inferring environmental information from data on large numbers of species, and present larger performance gains in this respect when the number of species increases.

**Qualitative Analysis** We conduct qualitative analysis by visualizing predictions for some random species (as shown in Figures 11 and 10). As shown, purely explicit and implicit models each have their own respective downsides. Implicit models generate vague predictions in the form of "blobs", with high scores at one or many center core(s) and lower scores spread out around the core(s). For instance, for Species D (*Agama lionotus*) the implicit model generated a

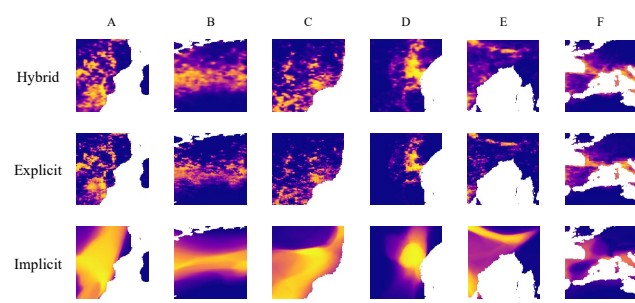

Figure 10: Selected local details of Figure 11.

prediction manifold centered at some place near the border between Kenya and Tanzania, spreading out into an egg-like shape. For Species E (*Monticola cinclorhyncha*) there seems to be two centers, respectively at South India and the Himalayas. This form of modeling seems to coincide with the well-known ecological Center-Periphery Hypothesis Pironon et al. (2017), which is a general rule-of-thumb but by no means an accurate pattern.

In comparison, explicit models create clusters of observation peaks that are disconnected from each other. This sometimes prompts more precise results compared to their implicit counterparts. For instance, Species A (*Merops bullockoides*) resides in savannahs BirdLife (2024). In southern Africa, savannahs exist in regions in and around Eswatini and Lesotho, but not further west into South Africa. This is represented well by the explicit model, while the implicit model's prediction reaches all the way to Cape Town. However, those clusters of prediction by the explicit models contain a lot of artifacts (*e.g.*, the obviously unnatural L-shaped border in southwestern France for Species F (*Clathus ruber*) predicted by the explicit model).

Our hybrid model combines those advantages. There are no blatant artifacts in our model, and the local clusters are more connected compared to the explicit model. There are also no large vague "blobs", *cf.* the implicit model. Thus, the results confirm our hybrid paradigm's superiority.

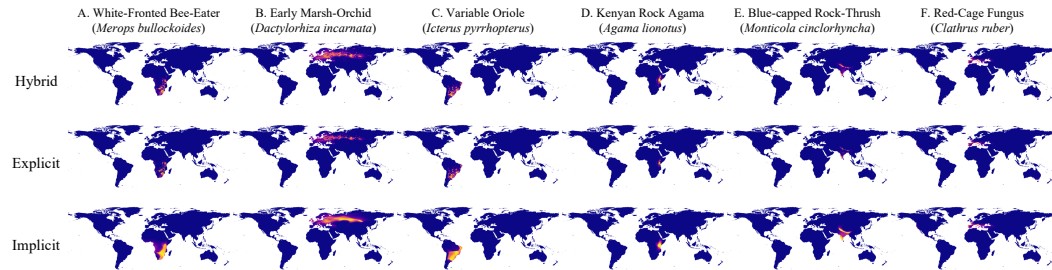

Figure 11: Visualizations of predictions for some randomly selected species. As shown, the hybrid model balances advantages of explicit and implicit representations: it predicts a continuous and integral range (instead of a shredded manifold like the result of the explicit model), while also giving regional details (instead of outputting a general range like the implicit model).

Table 4: Experiment with environmental data added.

| Implicitness | | 0.5 | 0.625 | 0.75 | 0.875 | SINR |
|---|---|---|---|---|---|---|
| S&T | 10 | 67.56 | 67.71 | 67.94 | 67.45 | 67.12 |
| | 100 | 77.10 | 77.23 | 77.08 | 77.03 | 76.88 |
| | 1000 | 79.85 | 80.26 | 80.49 | 80.23 | 80.48 |
| IUCN | 10 | 67.36 | 68.01 | 68.53 | 67.52 | 62.99 |
| | 100 | 76.37 | 76.64 | 76.76 | 76.26 | 74.49 |
| | 1000 | 76.55 | 77.10 | 76.21 | 75.89 | 76.07 |

**Ablation on Design Choices** To justify our use of sin-cos encoding and an implicit backbone based on FCNet, we compare with Fourier feature encoding (Tancik et al., 2020) and SIREN (Sitzmann et al., 2020). Results are shown in Table 5. As shown, FCNet and sin-cos encoding are both best-suited for the SDM task.

**Experiment with Environmental Data** In order to further investigate the relationship between our explicit embeddings and environmental information, we experiment with the same benchmarking task with environmental data from Cole et al. (2023). Results are shown in Table 4. As shown, we still achieve improvements; and in particular, improvements on S&T are marginal, whereas those on IUCN are significant. We attribute this to S&T being a simpler, easily saturated task, while IUCN presents a far greater challenge due to its larger number of species classes, higher weighting for data-sparse regions such as Africa, and more precise range requirements compared to the broader bird data in S&T. Notably, our model excels in sparse data settings (e.g., endangered species conservation), as evidenced by the strong performance on the observation cap 10 task. This shows hashgrids contain information complementary to environmental data, instead of just acting as a proxy. Also, on IUCN, our model with observation cap 100 is better than SINR with observation cap 1000, showing that we use data much more efficiently.

## 5 CONCLUSION

Our work explores the application of explicit and hybrid representations to the task of SDM construction. We introduce an innovative explicit representation scheme, and use it in conjunction with conventional implicit methods to form a hybrid representation. Experiments show that our hybrid models consistently achieve state-of-the-art accuracy on multiple standard benchmarks, outperforming both implicit and explicit models, and that our explicit representations are good at representing local environmental information. We also conducted extensive experiments to characterize our models and investigate their properties.

Table 5: Comparison with Fourier and SIREN.

| Method | Encoding | S&T | | | IUCN | | |
|---|---|---|---|---|---|---|---|
| | | 10 | 100 | 1000 | 10 | 100 | 1000 |
| Ours | sincos | 70.32 | 75.27 | 78.47 | 59.39 | 69.57 | 70.32 |
| FCNet | sincos | 65.59 | 73.12 | 76.81 | 50.98 | 62.06 | 65.57 |
| FCNet | Fourier | 65.36 | 72.94 | 76.09 | 50.48 | 68.54 | 69.96 |
| SIREN | sincos | 63.64 | 71.24 | 75.10 | 52.89 | 65.32 | 68.38 |
| SIREN | Fourier | 64.86 | 72.78 | 77.27 | 50.44 | 60.47 | 67.13 |

## ETHICS STATEMENT

Our work, like most similar ones on SDMs, is prone to the ethical hazards of damaging conservation fairness (Donaldson et al., 2016; Fedriani et al., 2017), insufficient reliability (Lee-Yaw et al., 2022), and potential for unintended uses such as poaching (Atlas & Dando, 2006). We suggest judicious use of our methods and careful interpretation of results.

## REPRODUCIBILITY STATEMENT

We open-sourced our implementation as a codebase, allowing all of our experimental results to be easily reproducible and making it easy for others to extend upon our work. We also released a zoo of all trained models under different learning rates, implicitness, and observation caps, such that the experiments can be repeated with minimal difficulty. Note that while Anonymous GitHub sometimes shows "The requested file is not found," this does not seem to affect downloading the repository.

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

## A  FULL RESULTS AND RAW DATA

**Benchmarking with Different Implicitness Settings**    Shown in Tables 4-6.

**Training and Inference Time**    Shown in Tables 7 and 8.

**Searching for Optimal Learning Rate**    Shown in Tables 9-11.

## B  MORE DESIGN INFORMATION

Across all experiments in our paper, the hashtable contained 524,288 values (2.10 MB). This is highly compressed compared to the 356 MB of environmental data — using less explicit values helps us avoid excessive training overhead, as each feature requires backprop, which is why our hybrid model can have no training or inference overhead compared to the implicit version. Additionally, the finest grid resolution used was 0.70º latitude × 1.41º longitude (77.9 km × 158.0 km near the equator) — we do not use a finer resolution because that would increase the amount of hash collisions. For forming the explicit grid, we use a Cartesian grid under the plate carrée projection.

Table 6: Our hashgrid *v.s.* Spherical Harmonics (SH). Note SIREN+SH performs well on S&T but is poor at IUCN, showing it is bad at making predictions in data-sparse regions.

| Implicit Backbone | Explicit Embedding | S&T 10 | S&T 100 | S&T 1000 | IUCN 10 | IUCN 100 | IUCN 1000 |
|---|---|---|---|---|---|---|---|
| FCNet | Hashgrid | 70.32 | 75.27 | 78.47 | 59.39 | 69.57 | 70.32 |
| FCNet | SH | 64.70 | 74.78 | 78.06 | 43.58 | 55.26 | 69.73 |
| SIREN | SH | 67.22 | 75.74 | 78.56 | 7.74 | 11.02 | 12.36 |

Table 7: Full results on S&T benchmark.

| Implicitness | Obs. Cap 10 | Obs. Cap 100 | Obs. Cap 1000 |
|---|---|---|---|
| 0.0 | 60.21±0.22 | 71.23±0.09 | 76.01±0.05 |
| 0.125 | 65.69±0.34 | 74.93±0.21 | 77.61±0.15 |
| 0.25 | 66.76±0.31 | 75.05±0.27 | 77.86±0.16 |
| 0.375 | 66.78±0.20 | 75.23±0.14 | 77.92±0.14 |
| 0.5 | 66.64±0.33 | 75.27±0.17 | 78.47±0.07 |
| 0.625 | 66.58±0.12 | 75.11±0.20 | 78.25±0.12 |
| 0.75 | 66.54±0.23 | 75.01±0.16 | 78.01±0.18 |
| 0.875 | 66.52±0.09 | 74.38±0.24 | 77.69±0.04 |
| 1.0 | 65.59±0.17 | 73.12±0.19 | 76.81±0.26 |

Table 8: Full results on IUCN benchmark.

| Obs. Cap | 0.0 | 0.125 | 0.25 | 0.375 | 0.5 | 0.625 | 0.75 | 0.875 | 1.0 |
|---|---|---|---|---|---|---|---|---|---|
| 10 | 48.83 | 56.39 | 58.30 | 59.56 | 59.39 | 58.29 | 58.28 | 57.52 | 50.98 |
| 100 | 62.42 | 66.71 | 69.02 | 69.41 | 69.57 | 68.95 | 69.23 | 67.69 | 62.06 |
| 1000 | 64.23 | 67.57 | 68.36 | 69.13 | 70.32 | 70.27 | 69.46 | 68.27 | 65.57 |

Table 9: Full results on GeoFeatures benchmark.

| Obs. Cap | 0.0 | 0.125 | 0.25 | 0.375 | 0.5 | 0.625 | 0.75 | 0.875 | 1.0 |
|---|---|---|---|---|---|---|---|---|---|
| 10 | 74.56 | 74.65 | 74.51 | 74.22 | 73.52 | 72.98 | 72.54 | 72.58 | 71.12 |
| 100 | 78.04 | 78.11 | 77.95 | 77.80 | 77.13 | 76.67 | 76.62 | 76.42 | 73.92 |
| 1000 | 79.28 | 79.19 | 79.03 | 78.87 | 78.59 | 78.12 | 77.88 | 77.48 | 75.17 |

Table 10: Full results on training time.

| Implicitness | Obs. Cap 10 | Obs. Cap 100 | Obs. Cap 1000 |
|---|---|---|---|
| 0.0 | 26.77±0.04 | 237.99±0.27 | 843.79±1.31 |
| 0.125 | 25.71±0.07 | 230.46±0.62 | 816.49±1.49 |
| 0.25 | 24.79±0.19 | 220.90±0.59 | 781.41±1.77 |
| 0.375 | 23.48±0.08 | 210.75±0.72 | 747.15±2.40 |
| 0.5 | 17.89±0.15 | 160.10±0.45 | 567.90±1.82 |
| 0.625 | 17.84±0.04 | 160.52±0.46 | 570.05±2.04 |
| 0.75 | 18.26±0.34 | 160.93±0.55 | 571.61±1.73 |
| 0.875 | 17.96±0.05 | 162.32±0.64 | 573.93±1.10 |
| 1.0 | 18.30±0.16 | 162.71±1.10 | 574.37±2.34 |

Table 11: Full results on inference time.

| Implicitness | Time (s) |
|---|---|
| 0.0 | 17.21±1.09 |
| 0.125 | 16.84±1.58 |
| 0.25 | 16.72±0.88 |
| 0.375 | 16.43±0.54 |
| 0.5 | 16.74±0.94 |
| 0.625 | 16.24±0.40 |
| 0.75 | 16.69±1.18 |
| 0.875 | 16.15±0.25 |
| 1.0 | 16.78±0.53 |

Table 12: Full results on S&T benchmark with different learning rates.

| Obs. Cap | Learning Rate | Implicitness | | | | | | | | |
|---|---|---|---|---|---|---|---|---|---|---|
| | | 0.0 | 0.125 | 0.25 | 0.375 | 0.5 | 0.625 | 0.75 | 0.875 | 1.0 |
| 10 | 0.01 | 42.29 | 40.95 | 42.66 | 43.49 | 44.87 | 46.87 | 22.78 | 54.48 | 64.52 |
| | 0.003 | **60.07** | 58.05 | 59.09 | 60.30 | 59.98 | 60.51 | 62.51 | 63.80 | 64.92 |
| | 0.001 | 58.13 | **64.57** | **66.86** | **67.00** | **66.73** | **66.46** | **66.40** | **66.58** | **65.41** |
| | 0.0003 | 24.98 | 57.13 | 63.21 | 64.18 | 64.97 | 65.47 | 65.30 | 65.15 | 64.46 |
| | 0.0001 | 24.42 | 34.12 | 44.49 | 47.93 | 50.78 | 52.98 | 53.43 | 55.38 | 57.91 |
| 100 | 0.01 | 48.21 | 50.40 | 50.97 | 52.32 | 57.32 | 59.12 | 23.50 | 29.34 | 70.73 |
| | 0.003 | 58.77 | 60.87 | 61.55 | 63.26 | 65.73 | 67.78 | 69.00 | 71.39 | 72.53 |
| | 0.001 | 70.27 | 70.64 | 71.62 | 72.52 | 73.23 | 73.44 | 73.72 | 74.13 | **73.17** |
| | 0.0003 | **71.25** | **74.98** | **75.35** | **75.53** | **75.44** | **75.24** | **75.14** | **74.30** | 72.76 |
| | 0.0001 | 64.05 | 71.08 | 72.13 | 72.12 | 72.25 | 72.43 | 72.46 | 71.95 | 71.26 |
| 1000 | 0.01 | 60.21 | 62.17 | 63.54 | 64.71 | 68.24 | 69.33 | 19.28 | 19.28 | 71.75 |
| | 0.003 | 65.25 | 66.99 | 68.07 | 68.58 | 72.12 | 72.26 | 73.81 | 74.44 | 75.78 |
| | 0.001 | 72.20 | 73.52 | 74.07 | 74.29 | 76.16 | 76.03 | 75.98 | 76.63 | **77.10** |
| | 0.0003 | **76.02** | **77.44** | 77.38 | 77.88 | **78.40** | **78.15** | **77.98** | **77.81** | 76.60 |
| | 0.0001 | 74.83 | 60.21 | **78.08** | **77.91** | 78.13 | 77.91 | 77.56 | 76.89 | 75.41 |

Table 13: Full results on IUCN benchmark with different learning rates.

| Obs. Cap | Learning Rate | Implicitness | | | | | | | | |
|---|---|---|---|---|---|---|---|---|---|---|
| | | 0.0 | 0.125 | 0.25 | 0.375 | 0.5 | 0.625 | 0.75 | 0.875 | 1.0 |
| 10 | 0.01 | 33.60 | 32.71 | 34.31 | 35.70 | 36.00 | 37.63 | 0.86 | 44.39 | 47.71 |
| | 0.003 | **48.83** | 52.22 | 53.32 | 54.41 | 53.79 | 54.42 | 55.25 | 56.16 | 49.87 |
| | 0.001 | 38.41 | **56.39** | **58.30** | **59.56** | **59.39** | **58.29** | **58.28** | **57.52** | **50.98** |
| | 0.0003 | 4.82 | 34.12 | 46.36 | 50.72 | 53.53 | 54.02 | 54.28 | 52.99 | 46.85 |
| | 0.0001 | 1.18 | 8.02 | 15.12 | 19.53 | 24.58 | 31.42 | 31.24 | 31.82 | 31.85 |
| 100 | 0.01 | 32.51 | 34.96 | 36.98 | 38.44 | 43.77 | 45.82 | 0.88 | 0.85 | 55.89 |
| | 0.003 | 48.67 | 49.54 | 51.09 | 53.05 | 56.54 | 58.19 | 60.54 | 61.77 | 58.52 |
| | 0.001 | **62.42** | 64.27 | 64.79 | 65.49 | 67.33 | 67.38 | 67.12 | 66.44 | 60.54 |
| | 0.0003 | 61.91 | **66.71** | **69.02** | **69.41** | **69.57** | **68.95** | **69.23** | **67.69** | **62.06** |
| | 0.0001 | 46.58 | 59.67 | 63.88 | 64.98 | 65.41 | 65.26 | 64.87 | 63.39 | 58.77 |
| 1000 | 0.01 | 35.99 | 38.95 | 40.47 | 42.25 | 48.24 | 50.42 | 1.00 | 0.85 | 54.01 |
| | 0.003 | 44.36 | 46.05 | 48.04 | 49.62 | 54.64 | 56.57 | 58.19 | 60.05 | 59.31 |
| | 0.001 | 58.68 | 59.70 | 61.11 | 61.37 | 64.45 | 64.90 | 66.08 | 65.43 | 64.40 |
| | 0.0003 | **64.23** | **67.57** | 68.03 | 68.56 | **70.32** | **70.27** | **69.46** | **68.27** | **65.57** |
| | 0.0001 | 60.73 | 66.53 | **68.36** | **69.13** | 69.50 | 68.41 | 68.83 | 67.15 | 62.30 |

Table 14: Full results on GeoFeatures benchmark with different learning rates.

| Obs. Cap | Learning Rate | Implicitness | | | | | | | | |
|---|---|---|---|---|---|---|---|---|---|---|
| | | 0.0 | 0.125 | 0.25 | 0.375 | 0.5 | 0.625 | 0.75 | 0.875 | 1.0 |
| 10 | 0.01 | 73.54 | 73.64 | 72.99 | 73.14 | 72.65 | 72.47 | 55.48 | 70.19 | 69.78 |
| | 0.003 | **74.56** | **74.65** | 74.03 | **74.22** | **73.52** | **72.98** | 72.38 | **72.58** | **71.12** |
| | 0.001 | 73.73 | 74.43 | **74.51** | 73.50 | 73.28 | 72.69 | **72.54** | 71.34 | 70.76 |
| | 0.0003 | 70.10 | 71.86 | 72.64 | 72.57 | 72.68 | 72.09 | 71.85 | 70.73 | 69.83 |
| | 0.0001 | 69.98 | 70.65 | 70.64 | 70.26 | 70.80 | 70.64 | 68.94 | 69.89 | 67.59 |
| 100 | 0.01 | 73.22 | 73.37 | 72.79 | 72.08 | 72.17 | 72.46 | 59.01 | 53.63 | 67.81 |
| | 0.003 | 75.90 | 76.61 | 75.54 | 75.65 | 76.00 | 74.54 | 75.12 | 74.28 | 71.01 |
| | 0.001 | **78.04** | 78.05 | 77.25 | 77.59 | **77.13** | 76.64 | **76.62** | 76.42 | 73.66 |
| | 0.0003 | 77.30 | **78.11** | **77.95** | **77.80** | 76.97 | **76.67** | 75.45 | 76.12 | **73.92** |
| | 0.0001 | 74.53 | 75.75 | 76.66 | 76.24 | 75.68 | 75.26 | 73.89 | 73.72 | 71.93 |
| 1000 | 0.01 | 74.10 | 74.98 | 74.50 | 74.93 | 74.32 | 74.45 | 62.69 | 53.68 | 62.99 |
| | 0.003 | 76.31 | 76.84 | 76.87 | 76.79 | 76.29 | 76.28 | 76.32 | 75.14 | 72.33 |
| | 0.001 | 78.98 | 78.75 | 78.74 | 78.52 | **78.59** | 77.36 | **77.88** | 77.44 | 74.82 |
| | 0.0003 | **79.28** | **79.19** | **79.03** | **78.87** | 78.32 | **78.12** | 77.69 | **77.48** | **75.17** |
| | 0.0001 | 78.80 | 78.45 | 78.91 | 78.58 | 78.42 | 77.89 | 77.17 | 76.80 | 74.18 |