# OpenReview forum: "Hybrid Spatial Representations for Species Distribution Modeling"
_ICLR.cc/2026/Conference — ICLR 2026 Conference Withdrawn Submission_

### Official Review · Reviewer_bDcE · 2025-10-25

**Soundness:** 2
**Presentation:** 2
**Contribution:** 2
**Rating:** 2
**Confidence:** 5

**Summary:**

This paper presents a hybrid spatial representation learning approach for species distribution modeling. Other than the neural implicit function-based location encoder, this paper also uses an explicit location encoding, which stores learnable location embeddings for individual regions and can be used to interpolate for any given regions.

**Strengths:**

1. The motivation section emphasize the problems of existing SDM models which helps the readers to understand the challenages.

2. The map visualization also helps readers to understand the problems.

**Weaknesses:**

The design of the model has significant problems that need to be addressed:

1. The explicit location encoding is not new. It was proposed a long time ago [1]. Its drawbacks have been studied in many previous studies [1, 2, 3]. There is an MAUP (Modifiable Areal Unit Problem). Which resolution level should the explicit location encoder store the learnable parameters? If the resolution is very small, the number of learnable parameters is very large and leads to overfitting. The cell without any training samples will not be trained. If the resolution is very large, the same smoothness problem will arise. The authors need to perform an ablation study on the picked resolution and report both the model performance as well as the number of learnable parameters.

2. The "low signal frequency" problem of implicit location encoders has been solved by recent multi-scale location encoders such as Space2Vec [1], Sphere2Vec [2], and SRNet [3]. The FCNet is an old model, and the state-of-the-art. Please compare with the recent location encoders [1,2,3].

3. There will be a map projection distortion problem when treating the lat/lon as 2D Euclidean coordinates. As shown by [2] and [3], this can significantly impact the model performance. This problem will be much worse for explicit location encoders such as grid cells in different latitudes correspond to different areas.

4. Moreover, for two points with longitude -179 and 179. They are very close on the Earth's surface, but treating lat/lon as 2D Euclidean coordinates makes them very far in the modeling space. I suggest using Google's S2 grid for the multiresolution grid instead of the current naive one.

4. There are too many technical problems in this paper. The authors seem not to be familiar with the existing literature and techniques, such as spatial index, location encoder, map projection, etc.

5. This hybrid location representation learning has been used in many tasks such as image synthesis [5] and satellite image super-resolution [6]. The authors need to conduct more literature reviews.

[1] Tang, Kevin, Manohar Paluri, Li Fei-Fei, Rob Fergus, and Lubomir Bourdev. "Improving image classification with location context." In Proceedings of the IEEE international conference on computer vision, pp. 1008-1016. 2015.

[2] Mai, Gengchen, Yao Xuan, Wenyun Zuo, Yutong He, Jiaming Song, Stefano Ermon, Krzysztof Janowicz, and Ni Lao. "Sphere2Vec: A general-purpose location representation learning over a spherical surface for large-scale geospatial predictions." ISPRS Journal of Photogrammetry and Remote Sensing 202 (2023): 439-462.

[3] Mai, Gengchen, Krzysztof Janowicz, Bo Yan, Rui Zhu, Ling Cai, and Ni Lao. "Multi-Scale Representation Learning for Spatial Feature Distributions using Grid Cells." In International Conference on Learning Representations.

[4] Rußwurm, Marc, Konstantin Klemmer, Esther Rolf, Robin Zbinden, and Devis Tuia. "Geographic Location Encoding with Spherical Harmonics and Sinusoidal Representation Networks." In The Twelfth International Conference on Learning Representations. 2024.

[5] Anokhin, Ivan, Kirill Demochkin, Taras Khakhulin, Gleb Sterkin, Victor Lempitsky, and Denis Korzhenkov. "Image generators with conditionally-independent pixel synthesis." In Proceedings of the IEEE/CVF conference on computer vision and pattern recognition, pp. 14278-14287. 2021.

[6] He, Yutong, Dingjie Wang, Nicholas Lai, William Zhang, Chenlin Meng, Marshall Burke, David Lobell, and Stefano Ermon. "Spatial-temporal super-resolution of satellite imagery via conditional pixel synthesis." Advances in Neural Information Processing Systems 34 (2021): 27903-27915.

**Questions:**

See the weakness.

---

### Official Review · Reviewer_3nVh · 2025-10-27

**Soundness:** 2
**Presentation:** 2
**Contribution:** 2
**Rating:** 4
**Confidence:** 3

**Summary:**

The authors propose a method for species distribution modelling (SDM). They focus mainly on models that are trained to provide a presence probability given a location (lat, long) on Earth, without the need for environmental variables (although they do explore that as well). Training is done on presence-only data and testing on expert made range maps. They argue that this can be done in two ways: (1) explicitly, by gridding the surface and directly learning the probability estimates of each grid cell and (2) implicitly, by learning a neural network that takes as input the coordinates and outputs the probabilities. The main contribution is showing that combining both approaches leads to better results than using each of them separately.

**Strengths:**

- The experimental results point towards an advantage of the proposed hybrid representation when compared to both the implicit and explicit representations.
- In particular, the proposed approach works better (or at least as well) than a large implicit model while running much faster.
- The paper is easy to follow and well written.

**Weaknesses:**

1. My main issue with this paper lies in its justification: (1) the argument is that implicit representations lead to parameters being shared across different locations (like between the Amazon and the Sahara). This is indeed the main advantage of such representations, since similar locations (in terms of species composition) will tend to share a similar representation; (2) a second argument is that implicit representations are unable to capture high frequency spatial patterns. Although it is true that they may lead to oversmoothness, this depends on the choice of location encodings and model size (as Table 3 suggests). After all, the explicit model can be obtained by feeding the implicit one with one-hot location encodings, meaning that the explicit model is a particular case of the implicit one.
2. The proposed approach for combining the two representations (by concatenating and varying the dimensionality of each) is unclear. According to Eq. (4) and the explanations around it, it seems like the implicit dimension $F_i$ and $M$ are kept constant, while $L$ is modified. According to line 242, $L$ may stand for both layers and levels, but it doesn’t show up in the sentence before Eq. (4). Are those levels related to the hashgrid scales? Since $F=F_i+L\times M$, for implicitness $F_i / F$ to be close to one, $L$ would have to be very large, I imagine leading to a very heavy hashgrid representation. Why not decreasing $F_i$ instead? What is the value of $L$ for the purely explicit model?
3. In Table 4, I assume the explicit model is the same as in the other experiments and the implicit one is substituted by an MLP that takes environmental variables as input (would be nice to clarify this). For reference, it would be important to add here also the explicitness 0 and 1 models (so the env MLP and the explicit model, which would be the same one as in the other table).
4. If I got it well, the Implicit line in Fig 8 should correspond to Fig 5 in the SINR paper. However, there are some surprising differences. Although capping at 100 seems to produce very similar results to SINR, capping at 10 seems to result in substantially worse performance (50 vs 60 MAP) than in the original paper. Similarly, capping at 1000 seems to always benefit from increasing the number of species in the original paper, but not in the present manuscript. I understand there’s quite a bit of stochasticity in the process (such as in the choice of the added species), but then it would be important to run these experiments multiple times with different random seeds.

**Questions:**

Each question relates to the corresponding weakness.
1. Related to weakness 1, my intuition is that the advantage of the proposed model is that it allows to efficiently increase the number of model parameters. After all, the implicit model tends to the hybrid model performance when its capacity is increased (Table 3). Could the authors explore this by comparing their approach with the implicit model in terms of performance vs number of parameters? I would find this to be a much more convincing argument than the current one.
2. I would like to see a clarified notation of the method, since that would allow me to ensure I have understood what the authors have done.
3. Could the authors provide the results for explicitness 0 and 1?
4. I understand that the time is short, but maybe the authors could provide a standard deviation for some of the cases (like in cap 10 with the fewest. species and cap 1000 with the most species).

---

### Official Review · Reviewer_AVaE · 2025-10-28

**Soundness:** 2
**Presentation:** 1
**Contribution:** 2
**Rating:** 2
**Confidence:** 4

**Summary:**

The authors propose to use an hybrid spatial representation approach for species distribution modeling (SDM). They focus on a setup where only location is provided to the model, and encode the location by combining both implicit and explicit embeddings. The latter is implemented with a multiresolution hashgrid. Their experiments show that having this hybrid combination is more effective than previous implicit only approaches, but also of fully explicit.

**Strengths:**

[S1] The proposed hybrid approach achieves notable performance gains on the Cole et al. (2023) dataset, indicating that integrating implicit and explicit representations is more effective than relying on a single representation in location-only SDMs.

[S2] The study provides a series of comparative results between explicit, implicit, and hybrid methods using the Cole et al. (2023) dataset.

**Weaknesses:**

[W1] Overall, I find that the paper shows a limited understanding of the SDM field, which results in weak motivation and unclear contributions.

- [W1.1] The introduction and related work sections are weak, and several statements are imprecise or incorrect. Some specific examples:
    - L37: Which models are you referring to here?
    - L50-53: This does not reflect the conclusions of Cole et al. (2023) and is inaccurate. High-quality data remain essential for obtaining precise SDMs. For instance, the GeoPlant dataset (Picek et al. (2024)) shows that presence–absence data are key to achieving top performance. What does “considerable accuracy” mean in this context? Models using environmental variables perform significantly better, consistent with ecological theory and SDM principles.
    - L67: This statement lacks supporting references. Moreover, its validity depends on the specific type of neural network, as some architectures address this issue.
    - L96: The claim that the state of the art is the FCNet architecture contradicts Russwurm et al. (2024), who show that SIREN networks tend to perform better. This incorrect statement is repeated at L172.
- [W1.2] The decision to omit environmental variables throughout the paper is conceptually flawed in my opinion. SDMs fundamentally rely on environmental variables because they capture the ecological drivers of species presence. Geographic coordinates can help model spatial autocorrelation not explained by environmental factors, but removing the latter simply weakens the model without clear justification. Specific points:
    - L75: Environmental data are not notoriously difficult to obtain or clean. Widely used datasets such as WorldClim are easy to access and substantially improve performance compared to location-only models.
    - L118: Selecting appropriate environmental variables does require domain expertise, but this is precisely because they are essential, not merely “helpful.”
- [W1.3] As a result, the paper’s motivation and claimed contributions remain unclear and unconvincing:
    - I do not see a real-world scenario where a location-only model would be preferable to one that incorporates relevant environmental variables. Even those used in Cole et al. (2023) (e.g., WorldClim and elevation) are insufficient to capture species distributions adequately. Recent work, such as Picek et al. (2024) with GeoPlant, confirms that richer, higher-resolution variables are essential. Consequently, experiments like those in Table 4, which include environmental variables, should be the base setup throughout the paper.
    - The proposed hybrid approach has already been explored in spatial modeling tasks, so its novelty lies only in its inappropriate application to SDMs. Comparisons with existing methods that integrate multi-scale information, such as the ones used in Russwurm et al. (2024), are largely missing. SIREN only appears in Table 5 and without Spherical Harmonics. It is also unclear how well these baselines were fine-tuned relative to the proposed approach (see [W2]).
    - In the abstract, two “main challenges” are mentioned — what exactly are they? The introduction lists challenges in bold, but the paper does not propose any new methodological solutions to address them, apart from the artificial challenge created by removing environmental data. The pipeline of Cole et al. (2023) appears to be reused almost entirely.
    - Based on the code, the only substantial change seems to be the use of the NeRF Studio implementation for explicit embeddings, which is not cited in Section 3.3. The lack of references there makes it unclear which parts of the method are original.

[W2] Nothing is said about hyperparameters, except at L353, where it is mentioned that “the reported results are under the best learning rate settings for each respective model.” Does this mean that hyperparameters were tuned on the test set? What about the other baselines, such as SIREN or the Fourier representation? Were these models fine-tuned as well to ensure a fair comparison?

[W3] Overall, the paper is not always clear, contains typos, and presents additional limitations such as:

- Figure 3 and its caption are not particularly clear. For example, the features are not “calculated”; they are learned, stored and then retrieved from the hashtable, correct?
- L188: I think this statement should be clarified or developed, as it is not obvious what the issue is. For me, parameters can be associated to multiple areas; it’s mainly a matter of how the embedding space is divided. Adding a reference could help.
- Typos: L149 – missing space; L174 – missing space; L366 – missing “of.”
- What does the asterisk in Table 2 refer to?
- The maps of Figure 10 are poorly informative: they appear almost random and do not reflect realistic species distributions. This likely stems from the arbitrary hybrid representation that excludes environmental variables. The analysis of these maps is also vague and inaccurate. For instance, why would the “obviously unnatural L-shaped border in southwestern France” be an artifact? Environmental conditions in southwestern France differ from those in the rest of the country, so differences in predictions are expected. Moreover, to help readers interpret these maps properly, you should include observed occurrences or expert range maps for comparison.

Picek, L., Botella, C., Servajean, M., Leblanc, C., Palard, R., Larcher, T., ... & Joly, A. (2024). Geoplant: Spatial plant species prediction dataset. *Advances in Neural Information Processing Systems*, *37*, 126653-126676.

**Questions:**

I think the framing of the paper needs to be reworked, and additional comparisons are required. I would therefore like the authors to:

[Q1] Provide a stronger justification for why environmental variables are not used, or alternatively, consider them as the base setup.

[Q2] Clarify which challenges the proposed method specifically addresses, and clearly state the paper’s main contributions.

[Q3] Include alternative baselines for encoding locations that tackle the same issues highlighted in the paper (basically the ones used in Russwurm et al. (2024))

[Q4] Explain how hyperparameters were selected and ensure that all baselines were given a fair opportunity to perform well.

[Q5] Address the line-specific comments, particularly those pointing out incorrect statements.

---

### Official Review · Reviewer_ssJL · 2025-11-01

**Soundness:** 3
**Presentation:** 2
**Contribution:** 3
**Rating:** 6
**Confidence:** 2

**Summary:**

This paper addresses the Species Distribution Modeling problem. Existing approaches typically use neural implicit representations, which often produce predictions with limited spatial precision in practice. This limitation is (partially) from their global formulation and inability to effectively capture local feature variations.

The authors instead combine an implicit location encoder with an explicit multiresolution hash-grid encoder to form a hybrid spatial representation. On the eBird S&T and IUCN benchmarks, the hybrid model achieves state-of-the-art performance, supported by ablations on implicitness, environmental correlation, efficiency, and qualitative map quality.

**Strengths:**

- Clear, convincing motivation for explicit + implicit: locality, non-Lipschitz boundaries, and parameter interpretability at grid cells.
- Limited data requirements, the authors use presence-only data (no presence–absence labels), require no additional environmental covariates, and rely on community-sourced datasets.
- Strong empirical gains on S&T and IUCN across data regimes.
- Explicit embeddings correlate more with environmental proxies (GeoFeatures), supporting the design hypothesis.
- Anonymous repo with trained model zoo are available.

**Weaknesses:**

1.	The paper mainly reports mAP and precision–recall curves. However, since this is a spatial modeling task, it should also measure how well the predicted ranges match the actual geographic boundaries.

2.	The model assumes that locations without observations are negative, but this may not always be true. I am not sure how these “pseudo-absences” affect the results.

3.	The results are averaged across all species, but it would be helpful to know which types of species or regions benefit most from the hybrid model.

4.	The writing is not very clear:

4.a.	Figure captions are not informative, especially Figure 2.

4.b.	In Figure 3, the line connections are hard to understand, they appear to start and end in odd places. I can’t clearly see the method being illustrated.

4.c.	In Section 3.1 Preliminaries, the writing is not well organized. It introduces many symbols at the beginning without first giving an intuitive explanation of the presence function. This function is essentially an indicator that maps a vector of geographic coordinates to the presence of each species. In addition, the distinction between $n$ and $N$ is confusing, and the “Assume Negative Full Loss” is used but not explained.

4.d.	In Section 3.2 Motivation, the argument based on existing global parameterization may be somewhat overstated. It is difficult to claim that the features of rainforests are completely isolated from those of deserts. The observed NIR limitations might instead result from the location embeddings themselves, where nearby locations share similar embeddings and thus overlook potentially significant differences. To support this claim, the authors should be more cautious and provide stronger evidence.

4.e.	The Method section feels unbalanced and not well introduced, I couldn’t clearly understand what the overall approach looks like at first. The motivation and Figures 4–5 take up too much space, and the figures are not referenced in the main text. Consider shortening this part and moving those figures to the appendix.

**Questions:**

- Since this is a spatial modeling task, can the authors report metrics that directly measure how close the predicted species ranges are to the ground truth boundaries (for example, overlap or distance-based metrics)?
- The paper assumes that locations without observations are negative. Is that a common choice across all other works? How sensitive are the results to different ways of selecting these pseudo-absences?
- The hash-table capacity is fixed in experiments. Does either hash-table size or collision rate being critical?
- Figure 1 appears less informative. Why does the explicit representation contain much more visible noise, and how can this “noise” be quantified or explained?

---

### Note · Authors · 2025-11-13

I have read and agree with the venue's withdrawal policy on behalf of myself and my co-authors.